# SnS_2_ as a Saturable Absorber for Mid-Infrared Q-Switched Er:SrF_2_ Laser

**DOI:** 10.3390/nano13131989

**Published:** 2023-06-30

**Authors:** Chun Li, Qi Yang, Yuqian Zu, Syed Zaheer Ud Din, Yu Yue, Ruizhan Zhai, Zhongqing Jia

**Affiliations:** 1International School for Optoelectronic Engineering, Qilu University of Technology (Shandong Academy of Sciences), Jinan 250353, China; lichun@qlu.edu.cn (C.L.); zuyuqian@qlu.edu.cn (Y.Z.); zaheer@qlu.edu.cn (S.Z.U.D.); zrz@vip.sdlaser.cn (R.Z.); jiazhongqing@vip.sdlaser.cn (Z.J.); 2School of Science, Shandong Jianzhu University, Jinan 250101, China; yueyu19@sdjzu.edu.cn

**Keywords:** 2D SnS_2_, TMDs, non-linear optical materials, mid-infrared laser

## Abstract

Two-dimensional (2D) materials own unique band structures and excellent optoelectronic properties and have attracted wide attention in photonics. Tin disulfide (SnS_2_), a member of group IV-VI transition metal dichalcogenides (TMDs), possesses good environmental optimization, oxidation resistance, and thermal stability, making it more competitive in application. By using the intensity-dependent transmission experiment, the saturable absorption properties of the SnS_2_ nanosheet nearly at 3 μm waveband were characterized by a high modulation depth of 32.26%. Therefore, a few-layer SnS_2_ was used as a saturable absorber (SA) for a bulk Er:SrF_2_ laser to research its optical properties. When the average output power was 140 mW, the passively Q-switched laser achieved the shortest pulse width at 480 ns, the optimal single pulse energy at 3.78 µJ, and the highest peak power at 7.88 W. The results of the passively Q-switched laser revealed that few-layer SnS_2_ had an admirable non-linear optical response at near 3 μm mid-infrared solid-state laser.

## 1. Introduction

Mid-infrared lasers have numerous practical applications due to their ability to cover multiple atmospheric windows, their capacity for strong absorption of a variety of molecules, and their ability to effectively concentrate thermal radiation energy [1,2]. Particularly, lasers with 3 μm wavelengths possess a superior capability when it comes to water absorption compared to other mid-infrared lasers [3]. K. S. Bagdasarov et al. produced the first 2.94 μm laser output at room temperature using an Er:YAG crystal in 1983, which paved the way for the study of the 3 μm laser [4]. According to research, highly doped Er ions are necessary to generate an efficient laser output when YAG-like oxides are used as the laser hosts [5,6]. However, it is challenging to generate crystals with high doping concentrations due to the limitations of the growing techniques. Fortunately, Er:SrF_2_ crystals, with low doping concentrations, have a high specific capacity to generate a 3 μm laser, which could avoid the difficulty of highly doping Er:YAG materials. In this study, bulk Er:SrF_2_ crystals were adopted as the gain material which was grown by the simple and cost-effective temperature gradient technique (TGT). And the doping concentration of Er ions was only 3%. Meanwhile, Er:SrF_2_ crystals, as the gain material, have excellent optical, mechanical, and thermal properties [6,7].

Passively Q-switched lasers could generate high-energy pulses up to several milli-joules with a simple laser structure. They gained significant applications in scientific research and medical treatment. Recently, researchers have shown that two-dimensional (2D) materials have the capacity to act as excellent saturable absorber (SA) materials, which further enhances the potential development and application prospects of pulse lasers [8,9,10,11,12]. Owing to their strong interaction with light, relatively high-charge carrier mobilities, exotic electronic properties, and excellent mechanical characteristics, transition metal dichalcogenides (TMDs) have attracted in-depth investigation and developed extraordinary applications [13,14,15]. Especially in the laser field, TMDs commonly possess a large modulation depth which is advantageous for great Q-switched pulse lasers [16,17]. Until now, many kinds of 2D TMDs, such as MoS_2_, SnS_2_, SnSe_2_, ReS_2_, and MoTe_2_, have been developed to apply in pulse lasers as SA materials [17,18,19,20,21,22]. Among them, Tin disulfide (SnS_2_) belongs to the IV-VI group TMDs with the CdI_2_ crystal structure [21]. Two layers are combined through van der Waals forces, which makes it easy to prepare 2D structures. Furthermore, 2D SnS_2_ owns excellent optical and electrical properties, making it widely used in the fields of ultrafast photonics and lasers [22,23,24]. Moreover, SnS_2_ consists of Sn and S elements which are abundantly stored elements in nature, and they own excellent properties of environmental optimization, low-cost, and nontoxicity [25]. Furthermore, SnS_2_ has oxidation resistance and thermal stability, which is beneficial to improve the application stability of devices made of SnS_2_ materials [26].

Research has shown that SnS_2_ was a direct bandgap semiconductor having a value of 2.24 eV [27]. Based on the Planck formula, the bandgap determines the absorbed photons by SnS_2_ in the visible optical energy region [27]. However, SnS_2_ SA has been achieved in near-infrared lasers, which mainly focus on the Yb-doped and Er-doped fiber laser in the wavelength range of 1–2 μm [25,26,27,28]. Compared with fiber lasers, solid-state lasers have the advantages of high power, simple structure, and excellent efficiency. So far, whether 2D SnS_2_ could be used as a SA in 3 μm solid-state lasers has not been verified yet.

In this study, the passively Q-switched Er:SrF_2_ lasers have been investigated with the few-layer SnS_2_ nanosheets employed as SA, which possess near 3 μm saturable absorption characteristics. Based on few-layer SnS_2_ nanosheets, the Er:SrF_2_ pulse laser was investigated using three kinds of resonators. When the output mirror had a radius of 100 mm and transmission of 1%, the shortest pulse width was 480 ns with a repetition rate of 37 kHz. Under the maximum absorbed power of 2.87 W, the laser acquired a single pulse energy of 3.78 µJ and a peak power of 7.88 W. At the output mirror of a radius of 100 mm and transmission of 4%, the Q-switched laser obtained the shortest pulse width of 820 ns, the maximum average output power of 87 mW, the single pulse energy of 2.18 µJ, and the peak power of 2.65 W. While the output mirror had a radius of 200 mm and transmission of 1%, the maximum average output power, repetition rate, pulse width, single pulse energy, and peak power were 149 mW, 40 kHz, 760 ns, 3.73 µJ, and 4.90 W, respectively.

## 2. Characterization of SnS_2_ SA

The morphologies were observed with scanning electron microscopy (SEM) and transmission electron microscopy (TEM), shown in Figure 1. Figure 1a depicts the SEM image assembled by a large number of thick nanosheets. Therefore, SnS_2_ needs to be prepared into a thin layer structure. A total of 50 mg of SnS_2_ powder was added to a 10 mL centrifuge tube filled with alcohol. Through 24 h ultrasonic and 15 min centrifugal treatment, the resulting supernatant was dropped onto a YAG-substrate with a diameter of 12.7 mm and left to air dry for 12 h. After the ultrasonic exfoliation method, the few-layer nanosheet structures were confirmed, as shown in Figure 1b.

The Raman spectrum was characterized for the SnS_2_ sample in Figure 2a. The peaks that are highlighted at 316.65 cm^−1^ and 206.04 cm^−1^ are attributed to the A_1g_ and E_g_ intralayer modes [28]. The absorption spectrum is measured when the wavelength is changed from 1.8 µm to 3 µm. As seen in Figure 2b, the SnS_2_ nanosheets show a relatively low, flat, and broad absorption, which indicates that SnS_2_ nanosheets are a promising broadband optical SA. The high transmission, in the mid-infrared band, is approximately 80–85%.

The non-linear saturable absorption was characterized in an intensity-dependent transmission experiment using a homemade laser with a repetition rate of 1 kHz and a pulse width of 1 μs at the wavelength of about 2.7 μm. Based on the Formula (1),
(1)$$T\left(I\right)=1-T_{ns}-\Delta T*exp\left(-I/I_{sat}\right)$$
where *T*(*I*): transmission; *I*: input intensity of the 2.7 μm laser; *T_ns_*: non-saturable absorbance; Δ*T*: modulation depth; *I_sat_*: saturation intensity. The experimental data and the fitted function are shown in Figure 3. The non-saturable absorbance was 23.07%, and the saturation intensity was 0.56 mJ/mm^2^. Moreover, the modulation depth was high, up to 32.26%, which indicates that the SnS_2_ materials have the capability as SA for the 2.7 μm passively Q-switched laser.

## 3. Pulse Laser Experiments

The Er:SrF_2_ passively Q-switched laser experiments were pumped by a laser diode (LD), emitting a continuous wave (CW) at a central wavelength of 976 nm. The pump laser was coupled by a fiber with a 105 μm radius. Using an optics system of 1:1, the laser was converged to the Er:SrF_2_ crystal with dimensions 3 mm × 3 mm × 10 mm. The Er:SrF_2_ crystal was mounted on a Cu holder. And the Cu holder was cooled with water to remove the excess heat and thus reduce the thermal effect. Mirrors M1 and M2 formed the resonator of the laser (shown in Figure 4). M1, working as an input mirror, was a plane mirror. M2 was the output mirror with a different radius and transmission. Three pulse laser experiments were investigated, when M2 had a radius of 100 mm with a transmission of 1% (T = 1%, R = 100 mm), a radius of 100 mm with a transmission of 4% (T = 4%, R = 100 mm), and a radius of 200 mm with a transmission 1% (T = 1%, R = 200 mm), respectively. When M2 had a radius of 100 mm, the length between M1 and M2 was the same as 80 mm. The Er:SrF_2_ laser was first operated as a CW laser; then, the SnS_2_ sample was placed into the resonant cavity. After carefully adjusting M1, M2, and the SnS_2_ sample, the passively Q-switched lasers were constructed. At this time, the SnS_2_ SA was 32 mm away from M1, and the spot size on the SA was calculated to be about 145 μm. While the radius of M2 was 200 mm, the cavity length was changed to 180 mm. When SnS_2_ SA was located 40 mm away from M1 with a spot size of approximately 170 μm, the stable Q-switched laser was obtained.

After the SnS_2_ SA was employed in the laser cavity, the Er:SrF_2_ passively Q-switched laser was successfully set up. For three laser cavities (T = 1%, R = 100 mm; T = 4%, R = 100 mm; T = 1%, R = 200 mm), the average output power and the absorbed pump power, shown in Figure 5a, all had a linear relationship. And the slope efficiencies were 5.22%, 4.28%, and 5.22%, respectively. When the absorbed pump power was increased to 2.87 W, the maximum average output power was 140 mW, 87 mW, and 149 mW, respectively. Comparing different transmittances, the average output power and slope efficiency acquired at a transmittance of 1% were higher than those of 4%, and the threshold absorbed pump power was lower because higher transmittance caused more loss. And, as shown in Figure 5b, when the transmittance was 1%, the passively Q-switched lasers were both dual-wavelength, located at 2729 nm and 2747 nm. At a transmittance of 4%, the central wavelength was 2728 nm.

As the absorbed pump power was increased from 0.53 W to 2.87 W, the pulse repetition rate gradually increased. To the three Q-switched lasers ((T = 1%, R = 100 mm), (T = 4%, R = 100 mm), and (T = 1%, R = 200 mm)), the highest pulse repetition rates were 37 kHz, 40 kHz, and 40 kHz, respectively. The detailed change rule is shown in Figure 6a. As shown in Figure 6b, the pulse widths were reduced with the increase of absorbed pump power. Three Q-switched lasers obtained the minimum pulse widths of 480 ns, 820 ns, and 760 ns. Comparing the results, the difference in repetition rate was small. Under the same transmission of 1%, the compact cavity design contributed to the compression pulse width.

Under the absorbed pump power of 2.87 W, the pulse shape, using three laser cavities, is shown in Figure 7. As can be seen, the pulse train exhibits fine repeatability. The single pulse has a good Q-switched waveform at 480 ns, 820 ns, and 760 ns, respectively. Comparing the results of pulse width, the short cavity length and low transmittance result in higher intracavity power density, effectively compressing laser pulse width. When Er:SrF_2_ laser absorbed the pump power of 2.87 W, the experiment acquired the single pulse energy of 3.78 µJ, 2.18 µJ, and 3.73 µJ. And the peak power was 7.88 W, 2.65 W, and 4.90 W, respectively. To sum up, when the radius of M2 was 100 mm with a transmission of 1%, the Er:SrF_2_ passively Q-switched laser obtained superb results.

Table 1 lists the data of some experiments using 2D SA materials for 2.7–3 µm pulse lasers in recent years. Owing to the exotic photoelectric properties, many 2D materials, working as SAs, have been researched for mid-infrared lasers, such as graphene, Bi_2_Se_3_, MoS_2_, BP, and so on. Due to its wide variety, 2D TMDs materials have received widespread attention, like TiSe_2_, WSe_2_, and ReS_2_. In this paper, the SnS_2_, belonging to TMDs, has proved a suitable SA for all-solid-state 2.7 µm lasers, having promoted the development of mid-infrared lasers. Moreover, compared with the other 2D SAs, SnS_2_ owns a modulation depth of up to 32.26%, making it an excellent SA material for Q-switched lasers.

## 4. Conclusions

Through experimental research, 2D SnS_2_ has been proven to have exceptional saturable absorption characteristics in near 3 μm mid-infrared laser. According to the intensity-dependent non-linear optical absorption theory, non-saturable absorbance, modulation depth, and saturation intensity were 23.07%, 32.26%, and 0.56 mJ/mm^2^, respectively. The diode-pumped Er:SrF_2_ laser adopted a compact plane-concave cavity. The laser operation realized the highest pulse energy of 3.78 µJ and a pulse peak power of 7.88 W, when the maximum average output power was 140 mW, and the shortest pulse duration was 480 ns at a repetition rate of 37 kHz, under M2 with a radius of 100 mm and transmission of 1%. Employing M2 with a radius of 100 mm and transmission of 4%, the Q-switched laser obtained the shortest pulse width of 820 ns, single pulse energy of 2.18 µJ, and peak power of 2.65 W. Using an M2 with a radius of 200 mm and transmission of 1%, the maximum average output power, the single pulse energy, and the peak power of the Er:SrF_2_ pulse laser were 149 mW, 3.73 µJ, and 4.90 W. The experiment results demonstrated that SnS_2_, having a high modulation depth, could act as a SA of solid-state laser with nearly 3 μm, which improves the selectivity of mid-infrared SA.

## Figures and Tables

**Figure 1 nanomaterials-13-01989-f001:**
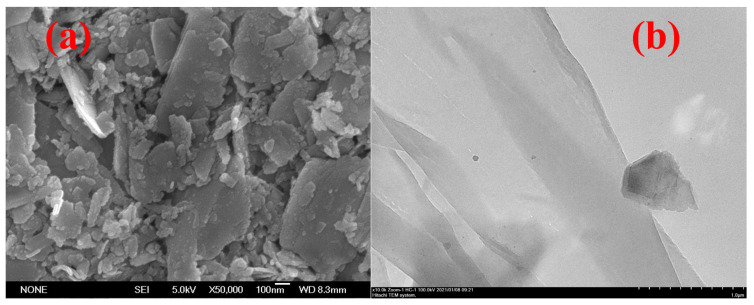
(**a**) SEM and (**b**) TEM images of the SnS_2_ sample.

**Figure 2 nanomaterials-13-01989-f002:**
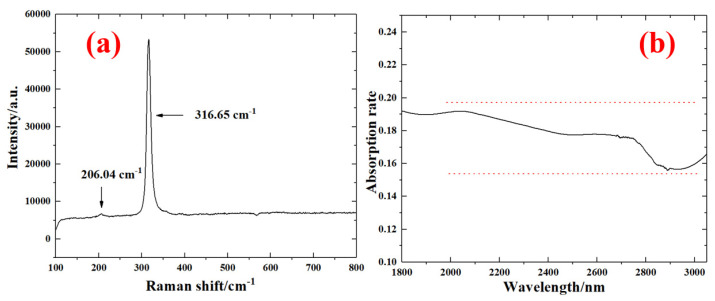
(**a**) The Raman spectrum and (**b**) absorption rate for the SnS_2_ sample.

**Figure 3 nanomaterials-13-01989-f003:**
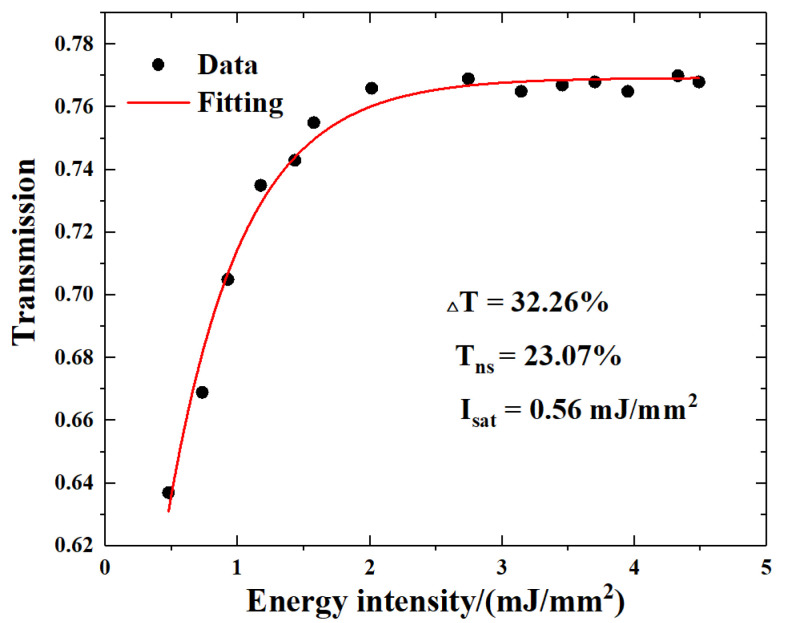
The function between transmission and energy intensity of SnS_2_ SA.

**Figure 4 nanomaterials-13-01989-f004:**
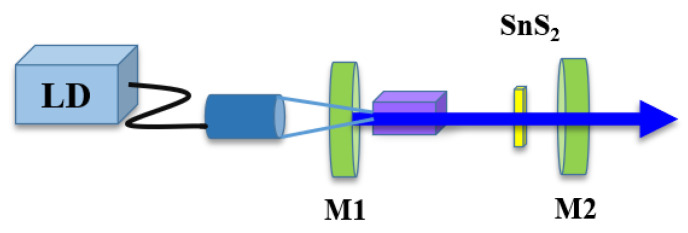
Experimental design of Er:SrF_2_ passively Q-switched laser.

**Figure 5 nanomaterials-13-01989-f005:**
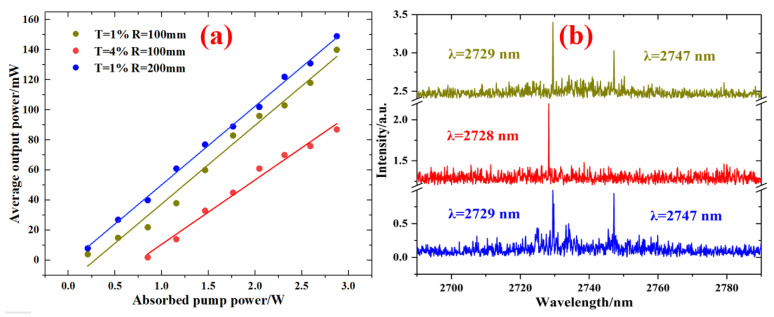
(**a**) Average output power versus absorbed pump power and (**b**) the Q-switched spectra.

**Figure 6 nanomaterials-13-01989-f006:**
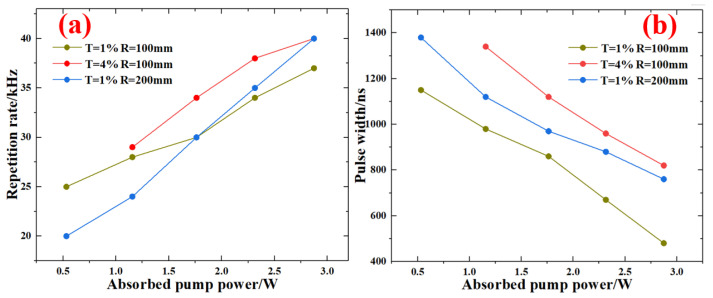
Repetition rate (**a**) and pulse width (**b**) as a function of absorbed pump power.

**Figure 7 nanomaterials-13-01989-f007:**
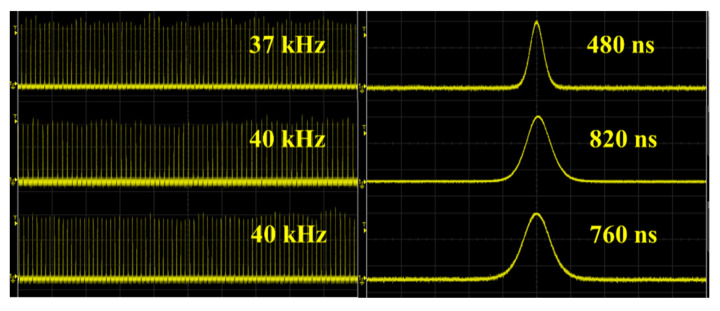
Pulse train and a single pulse of Er:SrF_2_ pulse laser at an absorbed pump power of 2.87 W.

**Table 1 nanomaterials-13-01989-t001:** Experimental results of bulk Q-switched lasers at near 3 µm by different 2D SAs.

SA	Gain	Modulation Depth /%	Output Power /mW	Pulse Width /ns	Repetition Rate /kHz	Pulse Energy /μJ	Wavelength/μm	Ref.
TiSe_2_	Ho,Pr:LLF	9.2	130	160.5	98.8	1.32	2.95	[29]
BP	Er:Lu_2_O_3_	8	755	359	107	7.1	2.84	[30]
TiC	Er:Lu_2_O_3_	5.5	896	266.8	136.9	6.5	2.85	[31]
WSe_2_	Er:Lu_2_O_3_	5.3	776	280	121	6.6	2.85	[32]
Nb_2_CT_x_	Er:Lu_2_O_3_	7.1	542	223.7	142.9	3.79	2.85	[33]
Ti_4_N_3_T_x_	Er:Lu_2_O_3_	8.6	778	278.4	113.7	6.84	2.85	[34]
MoS_2_	Er:Lu_2_O_3_	20.7	1030	335	121	8.5	2.84	[35]
ReSe_2_	Er:YAP	7.5	526	202.8	244.6	2.2	2.73 + 2.80	[36]
ReS_2_	Er:SrF_2_	3.8	580	508	49	12.1	2.79	[37]
Bi	Er:SrF_2_	1.82	226	980	56.2	4.02	2.73 + 2.75	[38]
SnS_2_	Er:SrF_2_	32.26	140	480	37	3.78	2.73 + 2.75	This work
SnS_2_	Er:SrF_2_	32.26	87	820	40	2.18	2.73	
SnS_2_	Er:SrF_2_	32.26	149	760	40	3.73	2.73 + 2.75	

## Data Availability

The data presented in this study are available upon request from the corresponding author.

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
