# Peer review of "SnS2 as a Saturable Absorber for Mid-Infrared Q-Switched Er:SrF2 Laser"

_nanomaterials, 2023, doi:10.3390/nano13131989_

Round 1

Reviewer 1 Report

The authors report on the use of few layers of 2D Tin disulfide (SnS2) as a saturable absorber (SA) for a bulk Er:SrF2 laser. They demonstrate passively Q-switched laser with the shortest pulse width of 480 ns and optimal single pulse energy was 3.78 μJ.

The paper is interesting for researchers working in the field of optoelectronics with 2D materials. A few points must be addressed before publication.

1.      More details regarding the laser cavity geometry must be reported, adding a photograph of the setup. How was the SnS2 inserted in the cavity? was it deposited onto a transparent window? And regarding the resonant mode in the cavity: what was the beam waist and the beam size on to the SnS2 film?

2.      What is the Q-switched mechanism operating with the SnS2? Is it active or passive Q-switching? And the pumping was continuous wave or pulsed?

3.      How many layers of SnS2 were used to obtain optimal Q-switched operation?

4.      What is the laser emission wavelength? It is not reported in the text. An output spectrum must be added

5.      In the comparison table the emission wavelength must be added for all the laser (line).

6.      The author must add a paragraph about the thermal stability under long term operation of their system.

7.      Figure 7 show an overmodulation of the pulse train (peak to peak variation of few percent). Can the authors comment on that?

Minor editing of English language

Reviewer 2 Report

Interesting paper on SnS2 research with functionality for the high-profile M-IR Q-sitche laser. Please further review below so that the reader can better understand exactly what is going on.

1) I don't understand the correlation between (a) and (b) in Fig. 1, which part of (a) becomes (b) if you peel off which part of (a) in the SEM and TEM images. Of course, I don't think you can find the absolute position in the pictures, but please explain the structural position. I don't understand how (a) doesn't look like an aggregate and becomes (b) on detachment.

2. The chapter on "3. Pulse laser experiments" is overloaded with information. It would be better to narrow down the necessary viewgraphs or show the data from several viewgraphs in a summarized and analyzed viewgraph. If you are going to show Figure 7, please show what the change in the width at half maximum of the peak means and what argument it brings to the discussion of this paper.

3. Table 1 is also just an example of information and comparison. Conclusions is not a technical report of a successful development, so please summarize the academic results.

Round 2

Reviewer 1 Report

The revision version of the paper contains all the requested revisions. It can be accepted in present form

minor editing

Author Response

We have thoroughly checked the manuscript and made the necessary revisions. We have diligently addressed typos, improved grammar, and ensured overall clarity. We appreciate the reviewer's feedback and the opportunity to enhance the manuscript. If there are any further concerns, we are open to addressing them accordingly.
